# Eigen-Distortions of Hierarchical Representations

**Alexander Berardino**
Center for Neural Science
New York University
agb313@nyu.edu

**Johannes Ballé**
Center for Neural Science
New York University*
johannes.balle@nyu.edu

**Valero Laparra**
Image Processing Laboratory
Universitat de València
valero.laparra@uv.es

**Eero Simoncelli**
Howard Hughes Medical Institute,
Center for Neural Science and
Courant Institute of Mathematical Sciences
New York University
eero.simoncelli@nyu.edu

## Abstract

We develop a method for comparing hierarchical image representations in terms of their ability to explain perceptual sensitivity in humans. Specifically, we utilize Fisher information to establish a model-derived prediction of sensitivity to local perturbations of an image. For a given image, we compute the eigenvectors of the Fisher information matrix with largest and smallest eigenvalues, corresponding to the model-predicted most- and least-noticeable image distortions, respectively. For human subjects, we then measure the amount of each distortion that can be reliably detected when added to the image. We use this method to test the ability of a variety of representations to mimic human perceptual sensitivity. We find that the early layers of VGG16, a deep neural network optimized for object recognition, provide a better match to human perception than later layers, and a better match than a 4-stage convolutional neural network (CNN) trained on a database of human ratings of distorted image quality. On the other hand, we find that simple models of early visual processing, incorporating one or more stages of local gain control, trained on the same database of distortion ratings, provide substantially better predictions of human sensitivity than either the CNN, or any combination of layers of VGG16.

Human capabilities for recognizing complex visual patterns are believed to arise through a cascade of transformations, implemented by neurons in successive stages in the visual system. Several recent studies have suggested that representations of deep convolutional neural networks trained for object recognition can predict activity in areas of the primate ventral visual stream better than models constructed explicitly for that purpose (Yamins et al. [2014], Khaligh-Razavi and Kriegeskorte [2014]). These results have inspired exploration of deep networks trained on object recognition as models of human perception, explicitly employing their representations as perceptual distortion metrics or loss functions (Hénaff and Simoncelli [2016], Johnson et al. [2016], Dosovitskiy and Brox [2016]).

On the other hand, several other studies have used synthesis techniques to generate images that indicate a profound mismatch between the sensitivity of these networks and that of human observers. Specifically, Szegedy et al. [2013] constructed image distortions, imperceptible to humans, that cause their networks to grossly misclassify objects. Similarly, Nguyen and Clune [2015] optimized randomly initialized images to achieve reliable recognition by a network, but found that the resulting

'fooling images' were uninterpretable by human viewers. Simpler networks, designed for texture classification and constrained to mimic the early visual system, do not exhibit such failures (Portilla and Simoncelli [2000]). These results have prompted efforts to understand why generalization failures of this type are so consistent across deep network architectures, and to develop more robust training methods to defend networks against attacks designed to exploit these weaknesses (Goodfellow et al. [2014]).

From the perspective of modeling human perception, these synthesis failures suggest that representational spaces within deep neural networks deviate significantly from those of humans, and that methods for comparing representational similarity, based on fixed object classes and discrete sampling of the representational space, are insufficient to expose these deviations. If we are going to use such networks as models for human perception, we need better methods of comparing model representations to human vision. Recent work has taken the first step in this direction, by analyzing deep networks' robustness to visual distortions on classification tasks, as well as the similarity of classification errors that humans and deep networks make in the presence of the same kind of distortion (Dodge and Karam [2017]).

Here, we aim to accomplish something in the same spirit, but rather than testing on a set of hand-selected examples, we develop a model-constrained synthesis method for generating targeted test stimuli that can be used to compare the layer-wise representational sensitivity of a model to human perceptual sensitivity. Utilizing Fisher information, we isolate the model-predicted most and least noticeable changes to an image. We test these predictions by determining how well human observers can discriminate these same changes. We apply this method to six layers of VGG16 (Simonyan and Zisserman [2015]), a deep convolutional neural network (CNN) trained to classify objects. We also apply the method to several models explicitly trained to predict human sensitivity to image distortions, including both a 4-stage generic CNN, an optimally-weighted version of VGG16, and a family of highly-structured models explicitly constructed to mimic the physiology of the early human visual system. Example images from the paper, as well as additional examples, are available at http://www.cns.nyu.edu/~lcv/eigendistortions/.

# 1   Predicting discrimination thresholds

Suppose we have a model for human visual representation, defined by conditional density $p(\vec{r}|\vec{x})$, where $\vec{x}$ is an $N$-dimensional vector containing the image pixels, and $\vec{r}$ is an $M$-dimensional random vector representing responses internal to the visual system (e.g., firing rates of a population of neurons). If the image is modified by the addition of a distortion vector, $\vec{x} + \alpha\hat{u}$, where $\hat{u}$ is a unit vector, and scalar $\alpha$ controls the amplitude of distortion, the model can be used to predict the threshold at which the distorted image can be reliably distinguished from the original image. Specifically, one can express a lower bound on the discrimination threshold in direction $\hat{u}$ for any observer or model that bases its judgments on $\vec{r}$ (Seriès et al. [2009]):

$$T(\hat{u}; \vec{x}) \geq \beta\sqrt{\hat{u}^T J^{-1}[\vec{x}]\hat{u}} \tag{1}$$

where $\beta$ is a scale factor that depends on the noise amplitude of the internal representation (as well as experimental conditions, when measuring discrimination thresholds of human observers), and $J[\vec{x}]$ is the Fisher information matrix (FIM; Fisher [1925]), a second-order expansion of the log likelihood:

$$J[\vec{x}] = \mathbb{E}_{\vec{r}|\vec{x}}\left[\left(\frac{\partial}{\partial\vec{x}}\log p(\vec{r}|\vec{x})\right)\left(\frac{\partial}{\partial\vec{x}}\log p(\vec{r}|\vec{x})\right)^T\right] \tag{2}$$

Here, we restrict ourselves to models that can be expressed as a deterministic (and differentiable) mapping from the input pixels to mean output response vector, $f(\vec{x})$, with additive white Gaussian noise in the response space. The log likelihood in this case reduces to a quadratic form:

$$\log p(\vec{r}|\vec{x}) = -\frac{1}{2}\left([\vec{r} - f(\vec{x})]^T[\vec{r} - f(\vec{x})]\right) + \text{const.}$$

Substituting this into Eq. (2) gives:

$$J[\vec{x}] = \frac{\partial f}{\partial\vec{x}}^T \frac{\partial f}{\partial\vec{x}}$$

Thus, for these models, the Fisher information matrix induces a locally adaptive Euclidean metric on the space of images, as specified by the Jacobian matrix, $\partial f/\partial\vec{x}$.

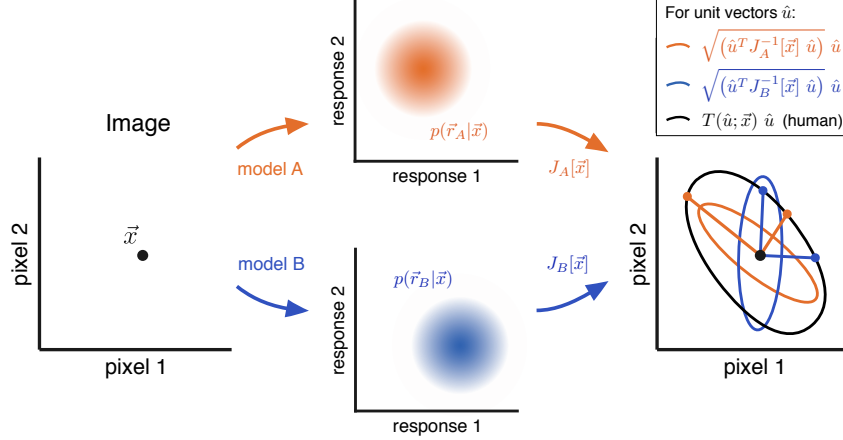

Figure 1: Measuring and comparing model-derived predictions of image discriminability. Two models are applied to an image (depicted as a point $\vec{x}$ in the space of pixel values), producing response vectors $\vec{r}_A$ and $\vec{r}_B$. Responses are assumed to be stochastic, and drawn from known distributions $p(\vec{r}_A|\vec{x})$ and $p(\vec{r}_B|\vec{x})$. The Fisher Information Matrices (FIM) of the models, $J_A[\vec{x}]$ and $J_B[\vec{x}]$, provide a quadratic approximation of the discriminability of distortions relative to an image (rightmost plot, colored ellipses). The extremal eigenvalues and eigenvectors of the FIMs (directions indicated by colored lines) provide predictions of the most and least visible distortions. We test these predictions by measuring human discriminability in these directions (colored points). In this example, the ratio of discriminability along the extremal eigenvectors is larger for model A than for model B, indicating that model A provides a better description of human perception of distortions (for this image).

## 1.1 Extremal eigen-distortions

The FIM is generally too large to be stored in memory or inverted. Even if we could store and invert it, the high dimensionality of input (pixel) space renders the set of possible distortions too large to test experimentally. We resolve both of these issues by restricting our consideration to the most- and least-noticeable distortion directions, corresponding to the eigenvectors of $J[\vec{x}]$ with largest and smallest eigenvalues, respectively. First, note that if a distortion direction $\hat{e}$ is an eigenvector of $J[\vec{x}]$ with associated eigenvalue $\lambda$, then it is also an eigenvector of $J^{-1}[\vec{x}]$ (with eigenvalue $1/\lambda$), since the FIM is symmetric and positive semi-definite. In this case, Eq. (1) becomes

$$T(\hat{e}; \vec{x}) \geq \beta/\sqrt{\lambda}$$

If human discrimination thresholds attain this bound, or are a constant multiple above it, then the ratio of discrimination thresholds along two different eigenvectors is the square root of the ratio of their associated eigenvalues. In this case, the strongest prediction arising from a given model is the ratio of the *extremal* (maximal and minimal) eigenvalues of its FIM, which can be compared to the ratio of human discrimination thresholds for distortions in the directions of the corresponding extremal eigenvectors (Fig. 1).

Although the FIM cannot be stored, it is straightforward to compute its product with an input vector (i.e., an image). Using this operation, we can solve for the extremal eigenvectors using the well-known power iteration method (von Mises and Pollaczek-Geiringer [1929]). Specifically, to obtain the maximal eigenvalue of a given function and its associated eigenvector ($\lambda_m$ and $\hat{e}_m$, respectively), we start with a vector consisting of white noise, $\hat{e}_m^{(0)}$, and then iteratively apply the FIM, renormalizing the resulting vector, until convergence:

$$\lambda_m^{(k+1)} = \left\| J[\vec{x}] \hat{e}_m^{(k)} \right\|; \quad \hat{e}_m^{(k+1)} = J[\vec{x}] \hat{e}_m^{(k)} / \lambda_m^{(k+1)}$$

To obtain the minimal eigenvector, $\hat{e}_l$, we perform a second iteration using the FIM with the maximal eigenvalue subtracted from the diagonal:

$$\lambda_l^{(k+1)} = \left\| (J[\vec{x}] - \lambda_m I) \hat{e}_l^{(k)} \right\|; \quad \hat{e}_l^{(k+1)} = (J[\vec{x}] - \lambda_m I) \hat{e}_l^{(k)} / \lambda_l^{(k+1)}$$

## 1.2 Measuring human discrimination thresholds

For each model under consideration, we synthesized extremal eigen-distortions for 6 images from the Kodak image set[2]. We then estimated human thresholds for detecting these distortions using a two-alternative forced-choice task. On each trial, subjects were shown (for one second each with a half second blank screen between images, and in randomized order) a photographic image (18 degrees across), $\vec{x}$, and the same image distorted using one of the extremal eigenvectors, $\vec{x} + \alpha \hat{e}$, and then asked to indicate which image appeared more distorted. This procedure was repeated for 120 trials for each distortion vector, $\hat{e}$, over a range of $\alpha$ values, with ordering chosen by a standard psychophysical staircase procedure. The proportion of correct responses, as a function of $\alpha$, was fit with a cumulative Gaussian function, and the subject's detection threshold, $T_s(\hat{e}; \vec{x})$ was estimated as the value of $\alpha$ for which the subject could distinguish the distorted image 75% of the time. We computed the natural logarithm of the ratio of these detection thresholds for the minimal and maximal eigenvectors, and averaged this over images (indexed by $i$) and subjects (indexed by $s$):

$$D(f) = \frac{1}{S}\frac{1}{I}\sum_{s=1}^{S}\sum_{i=1}^{I}\log\|T_s(\hat{e}_{li};\vec{x}_i)/T_s(\hat{e}_{mi};\vec{x}_i)\|$$

where $T_s$ indicates the threshold measured for human subject $s$. $D(f)$ provides a measure of a model's ability to predict human performance with respect to distortion detection: the ratio of thresholds for model-generated extremal distortions will be larger for models that are more similar to the human subjects (Fig. 1).

## 2 Probing representational sensitivity of VGG16 layers

We begin by examining discrimination predictions derived from the deep convolutional network known as VGG16, which has been previously studied in the context of perceptual sensitivity. Specifically, Johnson et al. [2016] trained a neural network to generate super-resolution images using the representation of an intermediate layer of VGG16 as a perceptual loss function, and showed that the images this network produced looked significantly better than images generated with simpler loss functions (e.g. pixel-domain mean squared error). Hénaff and Simoncelli [2016] used VGG16 as an image metric to synthesize minimal length paths (geodesics) between images modified by simple global transformations (rotation, dilation, etc.). The authors found that a modified version of the network produced geodesics that captured these global transformations well (as measured perceptually), especially in deeper layers. Implicit in both of these studies, and others like them (e.g., Dosovitskiy and Brox [2016]), is the idea that a deep neural network trained to recognize objects may exhibit additional human perceptual characteristics.

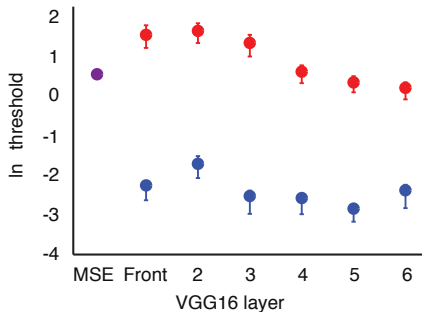

Figure 2: **Top:** Average log-thresholds for detection of the least-noticeable (red) and most-noticeable (blue) eigen-distortions derived from layers within VGG16 (10 observers), and a baseline model (MSE) for which distortions in all directions are equally visible.

Here, we compare VGG16's sensitivity to distortions directly to human perceptual sensitivity to the same distortions. We transformed luminance-valued images and distortion vectors to proper inputs for VGG16 following the preprocessing steps described in the original paper, and verified that our implementation replicated the published object recognition results. For human perceptual measurements, all images were transformed to produce the same luminance values on our calibrated display as those assumed by the model.

We computed eigen-distortions of VGG16 at 6 different layers: the rectified convolutional layer immediately prior to the first max-pooling operation (Front), as well as each subsequent layer following a pooling operation (Layer2–Layer6). A subset of these are shown, both in isolation and superimposed on the image from which they were derived, in Fig. 3. Note that the detectability of these distortions in isolation is not necessarily indicative of their detectability when superimposed

---

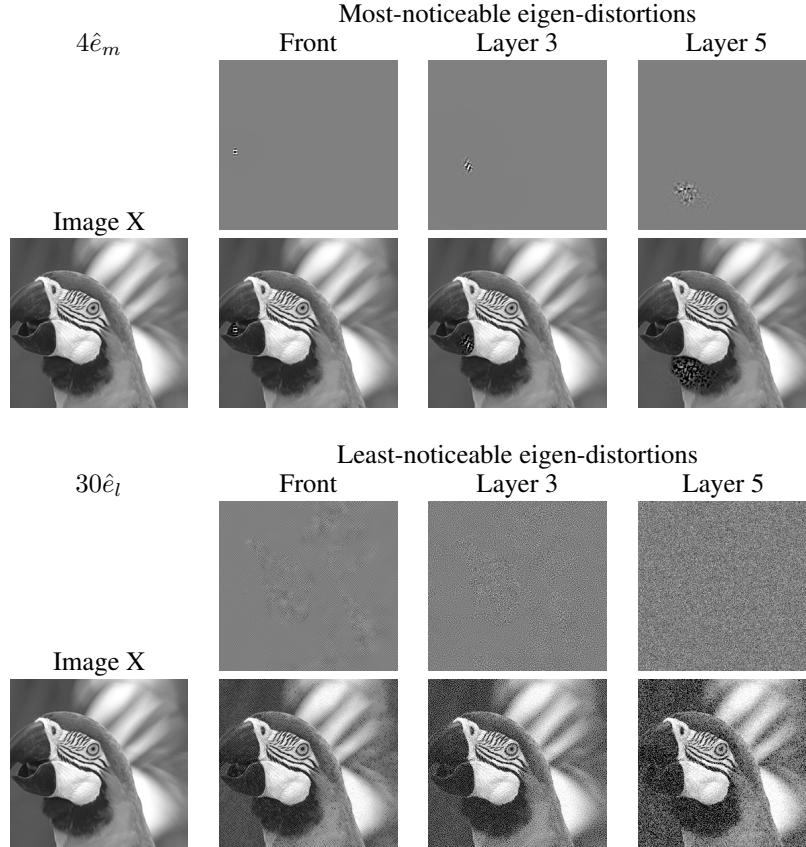

Figure 3: Eigen-distortions derived from three layers of the VGG16 network for an example image. Images are best viewed in a display with luminance range from 5 to 300 $cd/m^2$ and a $\gamma$ exponent of 2.4. **Top**: Most-noticeable eigen-distortions. All distortion image intensities are scaled by the same amount ($\times 4$). **Second row**: Original image ($\vec{x}$), and sum of this image with each of the eigen-distortions. **Third and fourth rows**: Same, for the least-noticeable eigen-distortions. Distortion image intensities are scaled the same ($\times 30$).

on the underlying image, as measured in our experiments. We compared all of these predictions to a baseline model (MSE), where the image transformation, $f(\vec{x})$, is replaced by the identity matrix. For this model, every distortion direction is equally discriminable, and distortions are generated as samples of Gaussian white noise.

Average Human detection thresholds measured across 10 subjects and 6 base images are summarized in Fig. 2, and indicate that the early layers of VGG16 (in particular, Front and Layer3) are better predictors of human sensitivity than the deeper layers (Layer4, Layer5, Layer6). Specifically, the most noticeable eigen-distortions from representations within VGG16 become more discriminable with depth, but so generally do the least-noticeable eigen-distortions. This discrepancy could arise from overlearned invariances, or invariances induced by network architecture (e.g. layer 6, the first stage in the network where the number of output coefficients falls below the number of input pixels, is an under-complete representation). Notably, including the "L2 pooling" modification suggested in Hénaff and Simoncelli [2016] did not significantly alter the visibility of eigen-distortions synthesized from VGG16 (images and data not shown).

## 3  Probing representational similarity of IQA-optimized models

The results above suggest that training a neural network to recognize objects imparts some ability to predict human sensitivity to distortions. However, we find that deeper layers of the network produce worse predictions than shallower layers. This could be a result of the mismatched training objective

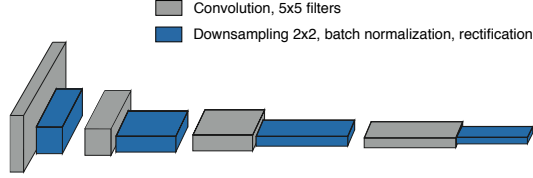

Figure 4: Architecture of a 4-layer Convolutional Neural Network (CNN). Each layer consists of a convolution, downsampling, and a rectifying nonlinearity (see text). The network was trained, using batch normalization, to maximize correlation with the TID-2008 database of human image distortion sensitivity.

function (object recognition) or the particular architecture of the network. Since we clearly cannot probe the entire space of networks that achieve good results on object recognition, we aim instead to probe a more general form of the latter question. Specifically, we train multiple models of differing architecture to predict human image quality ratings, and test their ability to generalize by measuring human sensitivity to their eigen-distortions.

We constructed a generic 4-layer convolutional neural network (CNN, 436908 parameters - Fig. 4). Within this network, each layer applies a bank of $5 \times 5$ convolution filters to the outputs of the previous layer (or, for the first layer, the input image). The convolution responses are subsampled by a factor of 2 along each spatial dimension (the number of filters at each layer is increased by the same factor to maintain a complete representation at each stage). Following each convolution, we employ batch normalization, in which all responses are divided by the standard deviation taken over all spatial positions and all layers, and over a batch of input images (Ioffe and Szegedy [2015]). Finally, outputs are rectified with a softplus nonlinearity, $\log(1 + \exp(x))$. After training, the batch normalization factors are fixed to the global mean and variance across the entire training set.

We compare our generic CNN to a model reflecting the structure and computations of the Lateral Geniculate Nucleus (LGN), the visual relay center of the Thalamus. Previous results indicate that such models can successfully mimic human judgments of image quality (Laparra et al. [2017]). The full model (On-Off), is constructed from a cascade of linear filtering, and nonlinear computational modules (local gain control and rectification). The first stage decomposes the image into two separate channels. Within each channel, the image is filtered by a difference-of-Gaussians (DoG) filter (2 parameters, controlling spatial size of the Gaussians - DoG filters in On and Off channels are assumed to be of opposite sign). Following this linear stage, the outputs are normalized by two sequential stages of gain control, a known property of LGN neurons (Mante et al. [2008]). Filter outputs are first normalized by a local measure of luminance (2 parameters, controlling filter size and amplitude), and subsequently by a local measure of contrast (2 parameters, again controlling size and amplitude). Finally, the outputs of each channel are rectified by a softplus nonlinearity, for a total of 12 model parameters. In order to evaluate the necessity of each structural element of this model, we also test three reduced sub-models, each trained on the same data (Fig. 5).

Finally, we compare both of these models to a version of VGG16 targeted at image quality assessment (VGG-IQA). This model computes the weighted mean squared error over all rectified convolutional layers of the VGG16 network (13 weight parameters in total), with weights trained on the same perceptual data as the other models.

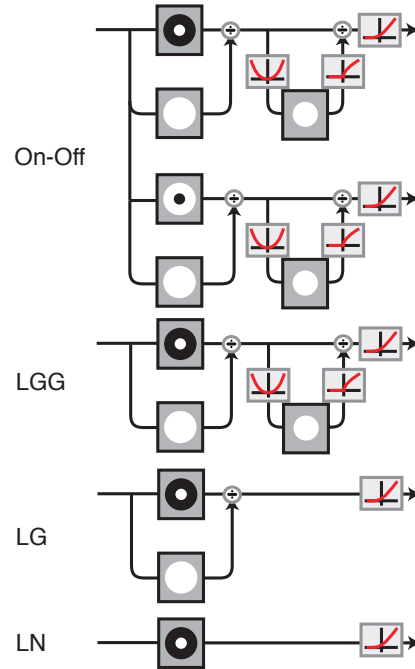

Figure 5: Architecture of our LGN model (On-Off), and several reduced models (LGG, LG, and LN). Each model was trained to maximize correlation with the TID-2008 database of human image distortion sensitivity.

## 3.1 Optimizing models for IQA

We trained all of the models on the TID-2008 database, which contains a large set of original and distorted images, along with corresponding human ratings of perceived distortion [Ponomarenko et al., 2009]. Perceptual distortion distance for each model was calculated as the Euclidean distance between the model's representations of the original and distorted images:

$$D_\phi = ||f_\phi(\vec{x}) - f_\phi(\vec{x}\,')||_2$$

For each model, we optimized its parameters, $\phi$, so as to maximize the correlation between the model-predicted perceptual distance, $D_\phi$ and the human mean opinion scores (MOS) reported in the TID-2008 database:

$$\phi^* = \arg\max_\phi \Big( \mathrm{corr}(D_\phi, MOS) \Big)$$

Optimization of VGG-IQA weights was performed using non-negative least squares. Optimization of all other models was performed using regularized stochastic gradient ascent with the Adam algorithm (Kingma and Ba [2015]).

## 3.2 Comparing perceptual predictions of generic and structured models

After training, we evaluated each model's predictive performance using traditional cross-validation methods on a held-out test set of the TID-2008 database. By this measure, all three models performed well (Pearson correlation: CNN $\rho = .86$, On-Off: $\rho = .82$, VGG-IQA: $\rho = .84$).

Stepping beyond the TID-2008 database, and using the more stringent eigen-distortion test, yielded a very different outcome (Figs. 7, 6 and 8). The average detection thresholds measured across 19 human subjects and 6 base images indicates that all of our models surpassed the baseline model in at least one of their predictions. However, the eigen-distortions derived from the generic CNN and VGG-IQA were significantly less predictive of human sensitivity than those derived from the On-Off model (Fig. 6) and, surprisingly, even somewhat less predictive than early layers of VGG16 (see Fig. 8). Thus, the eigen-distortion test reveals generalization failures in the CNN and VGG16 architectures that are not exposed by traditional methods of cross-validation. On the other hand, the models with architectures that mimic biology (On-Off, LGG, LG) are constrained in a way that enables better generalization.

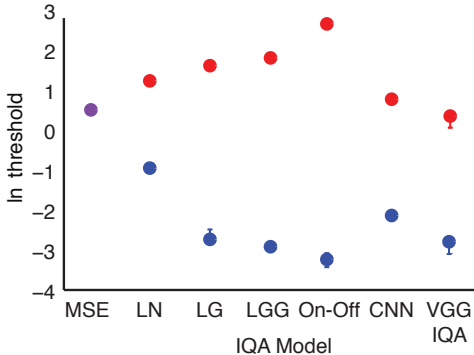

Figure 6: **Top:** Average log-thresholds for detection of the least-noticeable (red) and most-noticeable (blue) eigen-distortions derived from IQA models (19 human observers).

We compared these results to the performance of each of our reduced LGN models (Fig. 5), to determine the necessity of each structural element of the full On-Off model. As expected, the models incorporating more LGN functional elements performed better on a traditional cross-validation test, with the most complex of the reduced models (LGG) performing at the same level as On-Off and the CNN (LN: $\rho = .66$, LG: $\rho = .74$, LGG: $\rho = .83$). Likewise, models with more LGN functional elements produced eigen-distortions with increasing predictive accuracy (Fig. 6 and 8). It is worth noting that the three LGN models that incorporate some form of local gain control perform significantly better than the CNN and VGG-IQA models, and better than all layers of VGG16, including the early layers (see Fig. 8).

## 4 Discussion

We have presented a new methodology for synthesizing most and least-noticeable distortions from perceptual models, applied this methodology to a set of different models, and tested the resulting predictions by measuring their detectability by human subjects. We show that this methodology provides a powerful form of "Turing test": perceptual measurements on this limited set of model-optimized examples reveal failures that are not be apparent in measurements on a large set of hand-curated examples.

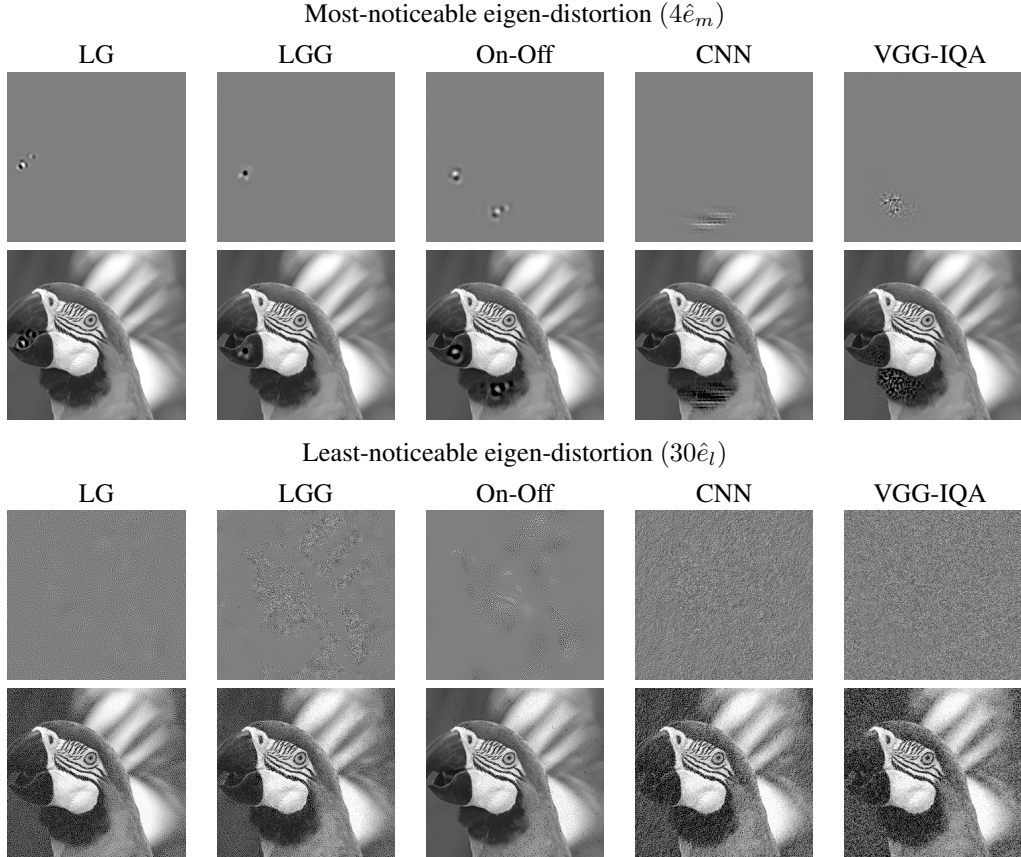

Figure 7: Eigen-distortions for several models trained to maximize correlation with human distortion ratings in TID-2008 [Ponomarenko et al., 2009]. Images are best viewed in a display with luminance range from 5 to 300 $cd/m^2$ and a $\gamma$ exponent of 2.4. **Top**: Most-noticeable eigen-distortions. All distortion image intensities are re-scaled by the same amount ($\times 4$). **Second row**: Original image ($\vec{x}$), and sum of this image with each eigen-distortion. **Third and fourth rows**: Same, for the least-noticeable eigen-distortions. All distortion image intensities re-scaled by the same amount ($\times 30$).

We are not the first to introduce a method of this kind. Wang and Simoncelli [2008] introduced Maximum Differentiation (MAD) competition, which creates images optimized for one metric while holding constant a competing metric's rating. Our method relies on a Fisher approximation to generate extremal perturbations, and uses the ratio of their empirically measured discrimination thresholds as an absolute measure of alignment to human sensitivity (as opposed to relative pairwise comparisons of model performance). Our method can easily be generalized to incorporate more physiologically realistic noise assumptions, such as Poisson noise, and could potentially be extended to include noise at each stage of a hierarchical model.

We've used this method to analyze the ability of VGG16, a deep convolutional neural network trained to recognize objects, to account for human perceptual sensitivity. First, we find that the early layers of the network are moderately successful in this regard. Second, these layers (Front, Layer 3) surpassed the predictive power of a generic shallow CNN explicitly trained to predict human perceptual sensitivity, but underperformed models of the LGN trained on the same objective. And third, perceptual sensitivity predictions synthesized from a layer of VGG16 decline in accuracy for deeper layers.

We also showed that a highly structured model of the LGN generates predictions that substantially surpass the predictive power of any individual layer of VGG16, as well as a version of VGG16 trained to fit human sensitivity data (VGG-IQA), or a generic 4-layer CNN trained on the same

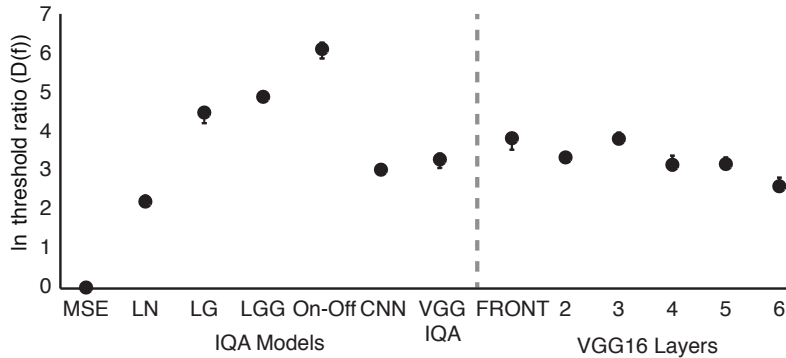

Figure 8: Average empirical log-threshold ratio (D) for eigen-distortions derived from each IQA optimized model and each layer of VGG16.

data. These failures of both the shallow and deep neural networks were not seen in traditional cross-validation tests on the human sensitivity data, but were revealed by measuring human sensitivity to model-synthesized eigen-distortions. Finally, we confirmed that known functional properties of the early visual system (On and Off pathways) and ubiquitous neural computations (local gain control, Carandini and Heeger [2012]) have a direct impact on perceptual sensitivity, a finding that is buttressed by several other published results (Malo et al. [2006], Lyu and Simoncelli [2008], Laparra et al. [2010, 2017], Ballé et al. [2017]).

Most importantly, we demonstrate the utility of prior knowledge in constraining the choice of models. Although the biologically structured models used components similar to generic CNNs, they had far fewer layers and their parameterization was highly restricted, thus allowing a far more limited family of transformations. Despite this, they outperformed the generic CNN and VGG models. These structural choices were informed by knowledge of primate visual physiology, and training on human perceptual data was used to determine parameters of the model that are either unknown or underconstrained by current experimental knowledge. Our results imply that this imposed structure serves as a powerful regularizer, enabling these models to generalize much better than generic unstructured networks.

## Acknowledgements

The authors would like to thank the members of the LCV and VNL groups at NYU, especially Olivier Henaff and Najib Majaj, for helpful feedback and comments on the manuscript. Additionally, we thank Rebecca Walton and Lydia Cassard for their tireless efforts in collecting the perceptual data presented here. This work was funded in part by the Howard Hughes Medical Institute, the NEI Visual Neuroscience Training Program and the Samuel J. and Joan B. Williamson Fellowship.

## Footnotes

*Currently at Google, Inc.

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
