[Reviews · NeurIPS 2017]

Reviewer 1



This is a great paper! And long overdue. The authors show that image distortions easily perceptible by humans are not well discriminated by neurons at higher levels of a deep net. This is done by a Fisher information analysis which computes how perturbations in pixel space translate into changes in the representation space at a given level of a deep net. Along with earlier findings showing that small image perturbations that are imperceptible to humans cause large changes in categorization, this study demonstrates that deep nets are not a good model of human visual perception, despite their impressive performance at trained object categorization tasks. It is an important neural network "psychophysics" study that should be of interest to NIPS. One question that arises from this study is whether the top layer of a neural net is the best or only level to tap into for obtaining information about the image. There is a common assumption that the higher levels of visual cortex correspond to the "top box" that is more closely tied to our perception, but increasing evidence points to the idea that *all* levels contribute information for solving a task, and which level is most important depends on the nature of the task. So in that sense the fact that lower levels are more closely tied to human subjects judgements of low level perturbations makes sense. But the fact that even a simple neurophysiolgoically informed front-end model outperforms the CNN networks seems to indicate that they are missing some fundamental components, or that the specific categorization task is driving them away from achieving a more generically useful image representation.

Reviewer 2



The submission presents a method to generate image distortions that are maximally/minimally discriminable in a certain image representation. The maximally/minimally distortion directions are defined as the eigenvectors of the Fisher Information Matrix with largest/smallest eigenvalue. Distortions are generated for image representations in the VGG-16 as well as for representations in models that were trained to predict human sensitivity to image distortions. Human discrimination thresholds for those distortions are measured. It is found that the difference in human discrimination threshold between max and min distortions of the model is largest for a biologically inspired 'early vision' model that was trained to predict human sensitivity, compared to a CNN trained to predict human sensitivity or the VGG-16 representations. For the VGG representations it is found that the difference in detection threshold for humans is larger for min/max distortions of earlier layers than for later layers. Thus it is concluded that the simple 'early vision' representation is better aligned with human perceptual representations than those of the VGG network trained on object recognition. Pro - The idea of generating extremal distortions of a certain representation seems appealing and a useful tool for visualising important aspects of complex image representations. - The finding that the max/min distortions of the early vision model are better aligned with human discrimination thresholds than those of the VGG is interesting and could be useful for the design of 'perceptual loss functions' in tasks like super-resolution. Contra - The extremal distortions are an informative but limited aspect of the image representations. I am not sure to what extend one can conclude from the results that one representation is in general closer to human perception than another. Comments - I would be very interested if the VGG representation improves if it is fine-tuned to predict human sensitivity to image distortions.

Reviewer 3



PAPER SUMMARY The paper attempts to measure the degree to which learned image representations from convolutional networks can explain human perceptual sensitivity to visual stimuli. This question is well-motivated on the neuroscience side by recent studies that have used learned image representations to predict brain activity in primates, and on the computer vision side by recent methods for image synthesis which have used image representations learned for object recognition as a surrogate for human perception. The paper develops a clever experimental paradigm to test the extent to which a (differentiable) image representation agrees with human perception by experimentally measuring the human discrimination threshold for image distortions along eigendirections of the Fisher information matrix corresponding to extremal eigenvalues. Using this experimental technique, the paper demonstrates that for a VGG-16 model pretrained for ImageNet classification, image representations at earlier layers correlate with human perception more strongly than those at later layers. The paper next explores image representations from models trained explicitly to match human judgements of perceived image distortion, training both a four-layer CNN and an On-Off model whose structure reflects the structure of the human LGN. Both models are able to fit the training data well, with the CNN slightly outperforming the On-Off model. However using the experimental paradigm from above, the paper shows that image representations from the On-Off model correlate much more strongly with human perception than either the shallow CNN model or representations from any layer of the VGG-16 model trained on ImageNet. FEATURES FROM IQA CNN For the four-layer IQA CNN from Section 3.1, it seems that only the image representation from the final layer of the network were compared with human perceptual judgements; however from Figure 3 we know that earlier layers of VGG-16 explain human perceptual judgements much better than later layers. I’m therefore curious about whether features from earlier layers of the IQA CNN would do a better job of explaining human perception than final layer features. RANDOM NETWORKS AND GANS Some very recent work has shown that features from shallow CNNs with no pooling and random filters can be used for texture synthesis nearly as well as features from deep networks pretrained on ImageNet (Ustyuzhaninov et al, ICLR 2017), suggesting that the network structure itself and not the task on which it was trained may be a key factor in determining the types of perceptual information its features capture. I wonder whether the experimental paradigm of this paper could be used to probe the perceptual effects of trained vs random networks? In a similar vein, I wonder whether features from the discriminator of a GAN would better match human perception than ImageNet models, since such a discriminator is explicitly trained to detect synthetic images? ON-OFF MODEL FOR IMAGE SYNTHESIS Given the relative success of the On-Off model at explaining human perceptual judgements, I wonder whether it could be used in place of VGG features for image synthesis tasks such as super-resolution or texture generation? OVERALL Overall I believe this to be a high-quality, well-written paper with an interesting technical approach and exciting experimental results which open up many interesting questions to be explored in future work. POST AUTHOR RESPONSE I did not have any major issues with the paper as submitted, and I thank the authors for entertaining some of my suggestions about potential future work. To me this paper is a clear accept.